# Application of Machine Learning Techniques to Predict the Mechanical Properties of Polyamide 2200 (PA12) in Additive Manufacturing

Ivanna Baturynska 

Department of Manufacturing and Civil Engineering, Norwegian University of Science and Technology, 2801 Gjovik, Norway; ivanna.baturynska@ntnu.no; Tel.: +47-477-32-882

**Abstract:** Additive manufacturing (AM) is an attractive technology for the manufacturing industry due to flexibility in its design and functionality, but inconsistency in quality is one of the major limitations preventing utilizing this technology for the production of end-use parts. The prediction of mechanical properties can be one of the possible ways to improve the repeatability of results. The part placement, part orientation, and STL model properties (number of mesh triangles, surface, and volume) are used to predict tensile modulus, nominal stress, and elongation at break for polyamide 2200 (also known as PA12). An EOS P395 polymer powder bed fusion system was used to fabricate 217 specimens in two identical builds (434 specimens in total). Prediction is performed for XYZ, XZY, ZYX, and Angle orientations separately, and all orientations together. The different non-linear models based on machine learning methods have higher prediction accuracy compared with linear regression models. Linear regression models only have prediction accuracy higher than 80% for Tensile Modulus and Elongation at break in Angle orientation. Since orientation-based modeling has low prediction accuracy due to a small number of data points and lack of information about the material properties, these models need to be improved in the future based on additional experimental work.

**Keywords:** additive manufacturing; machine learning; tensile modulus; predictive modeling; mechanical properties; polyamide 2200; PA12

## 1. Introduction

In polymer powder bed fusion additive manufacturing, anisotropic behavior of the material leads to the variation in dimensional and mechanical properties depending on the part orientation. Since this technology has obtained attention from automotive, aerospace, and medical industries as a technology for producing end-use parts, requirements for part quality have increased significantly, including the need for consistent results. Therefore, the development of new models for the prediction of mechanical properties will allow meeting these requirements.

The current state-of-the-art [1–19] describes the importance of part orientation, powder morphology, and machine process parameters as a means towards the control and management of variation in polymer powder bed fusion system. Among the most investigated additive manufacturing (AM) machine process parameters are laser power, scan speed, hatch distance, scan strategy, beam speed, melting temperature, and powder bed temperature [2–7]. There is a number of studies [20–22] which report that laser power, scan speed, hatch distance, and layer thickness can be used to define the line energy and how their variation may influence mechanical properties of the part. In addition to energy applied to solidify polyamide, Mielicki et al. [8] have also reported the importance of layer thickness, and powder distribution in each layer. Powder distribution is dependent on the size of the particles in the powder and powder viscosity.

Futhermore, Drummer et al. [18] and Gümüs et al. [23] studied how the size of the particles of the polymer powder and its viscosity influence mechanical properties. While Drummer et al. [18] investigated the degradation behavior of PA12 based on the analysis of phase transition temperature and melt viscosity for both virgin and aged powder, Gümüs et al. [23] have reported that the size and morphology of the particles could lead to the creation of pores, gaps, or/and voids in the fabricated parts. Thus, Flores Ituarte et al. [22] have proposed to use a machine vision in combination with Design of Experiment (DOE) to monitor and control powder morphology. In order to control powder flowability, the authors described how machine process parameters need to be varied (e.g., line energy, change the speed of the recoater, adjust layer thickness).

The influence of part orientation on mechanical properties has already been described in detail by [12,24,25]. Besides, there is a difference between what is reported in the literature and what is provided by EOS data sheets. While [12,24,25] have reported that Tensile modulus, Elongation at break, and Maximal Stress are affected by the part orientation, EOS reports in their data sheets for PA12-Balanced process parameters group that Tensile Modulus is the same in all orientations. Ituarte et al. [26] have also reported that part orientation has the most significant influence on mechanical properties among other investigated parameters based on the Taguchi DOE. Besides, Caulfield et al. [12] reported that the thermal distribution in the build chamber also has an impact on the mechanical properties of the fabricated parts.

However, there is a limited number of studies that have attempted to predict mechanical properties based on the part positioning in the build chamber. Similar research was performed for the investigation of dimensional accuracy for PA12 based on the part positioning in the build chamber, part orientation, and STL model properties (number of mesh triangles, volume, and surface) [11]. Using a similar strategy would contribute to the development of the schematic approach of positioning parts in the build chamber based on their requirements, and thus an increasing area of build chamber utilization, which would lead to cheaper and more sustainable production.

Therefore, in this study the main focus is set on the development of linear and non-linear models to predict Tensile Modulus, Nominal Stress, and Elongation at break by using different machine learning techniques. Machine learning techniques have already been used for prediction of geometrical deviations, and have shown a great potential for the datasets collected from more than 100 samples [27]. However, the application of the classical approaches like DOE for smaller batches of data has shown great potential and has been reported in [22,26,28].

While the structured DOE is beneficial for the optimization of individual manufacturing parameters of AM systems and allows developing a systematic experimental approach to simultaneously analyzing multiple production requirements [26], the structured DOE also has drawbacks. For example, Flores Ituarte et al. [22] have reported that "The presented DOE was incapable of replicating this phenomenon, as the range energy density was limited to a narrower window due to excessive geometrical distortion", which means that the investigation of different combinations of AM-related parameters depends on the understanding of the process, and in the case where some of the process conditions are not satisfied, the data from the experiment can be lost. Therefore, there is a need for an additional data analysis technique, which will be able to learn the correlations between parameters that cannot be seen with a "human eye" from previous studies.

Machine learning techniques have an ability to find hidden patterns in the data, and then accumulated information can be transformed to the new unknown problems [9]. In addition, machine learning techniques do not require a structured way of data collection, and thus they have a benefit over other classical methods when it comes to the conditions similar to real-time manufacturing. Data collected using a different design of experiments can be used together (joined in one data set) for model development, and with every next use of AM machines, collected data contributes to modification of previously developed models, resulting in higher prediction accuracy and better precision for the future use of AM. Nowadays, these methods are used in different field of studies [29–31], and have also been widely used in traditional manufacturing [32–35].

Specifically, the author aims to adress the following issues (Section 3):

- Estimation of mechanical properties of AM-manufactured parts without prior knowledge about the material.
- Understanding how the mechanical properties depend on the part positioning in the build chamber.
- Compare performance of Linear regression models and machine learning proposed models, and choose the best models for the prediction of mechanical properties.
- Discuss which of the investigated features are the most significant and can be used to predict the mechanical properties, and how the mechanical properties can be controlled and managed based on the obtained results.

## 2. Materials and Methods

Experimental work was performed on EOS P395 polymer powder bed fusion system with Polyamide 12 (PA12) used as a material. The PA12 powder was used with a 50%/50% ratio of virgin/self-aged powder, respectively, and both virgin and self-aged powder were taken from the same batch. The self-aging of powder was done based on the approach presented by Rüsenberg et al. [3]. However, part placement and orientation strategy (see Figure 1) were chosen to be different, and it is described in more detail in Section 2.1.

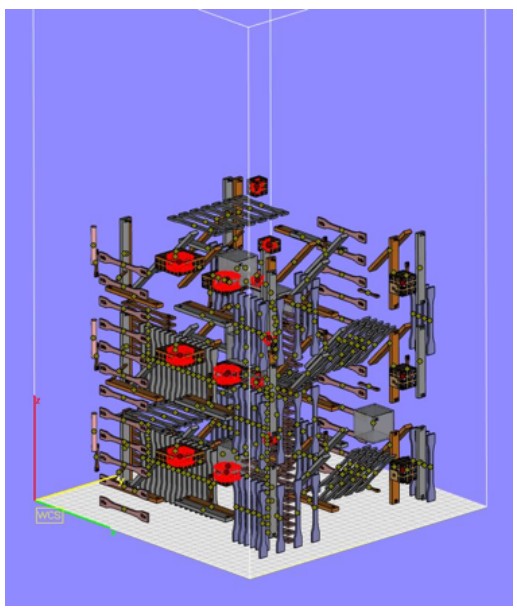

**Figure 1.** Build layout in Magics 20.0.

Figure 2 presents the main stages of the experiment, starting from aging powder and finishing with the execution of Tensile testing. At the first aging stage, 100% of virgin PA12 (40 kg) was used in the EOS P395 machine without energy deposition. The next two aging steps have used a powder only from the build cake (from build chamber) obtained as a result of the previous aging step. Since the amount of self-aged powder from Run 1 was not enough for the execution of two experimental builds, an additional run for aging powder was performed (Run 2 on Figure 2). Run 2 consisted of three similar aging steps as in Run 1, with the same process parameters. In total, 45 kg of self-aged Polyamide 12 was obtained and mixed with 45 kg of virgin powder. Then mixed powder was divided into two equal batches to be used in two experimental builds.

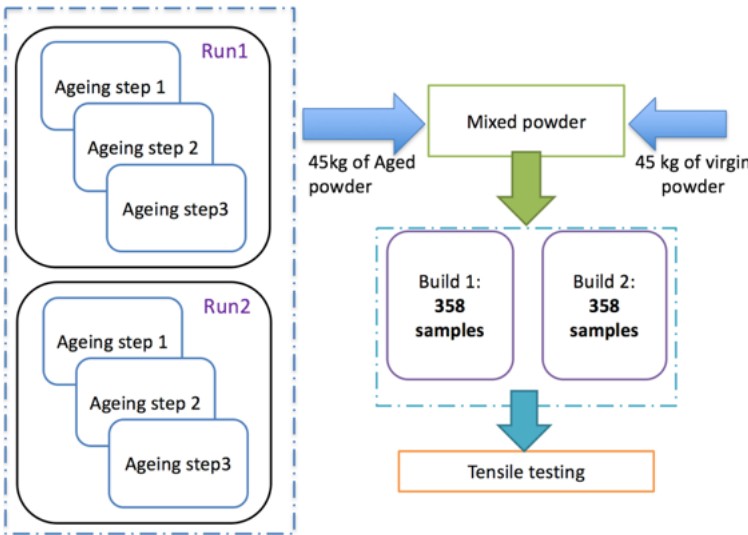

**Figure 2.** Schematic representation of main stages of the experiment.

Since in this experiment the idea of using a self-aged powder proposed by Rüsenberg et al. [3] has been utilized to control material properties, it was of interest to compare whether similar repeatability of the results will be obtained to the ones Rüsenberg et al. [3] presented in their work. Therefore, build layout, process, and material parameters were kept constant through the whole experiment and are shown in Table 1.

**Table 1.** Material and process parameters used in the experiment.

| Parameters | Value |
| --- | --- |
| Virgin/aged PA2200 powder ratio,% | 50/50 |
| Layer thickness, mm | 120 |
| EOS P395 system settings | Balance |
| AM system warm up time, min | 120 |
| AM system cooling down time, min | 240 |
| Working chamber temperature, °C | 180.5 |
| Removal chamber temperature, °C | 130.0 |

### 2.1. Build Layout

The work by Rüsenberg et al. [3] was taken as a reference work, but a number of specimens and placement strategies were altered. It was chosen to produce 217 standardized specimens of ISO 527-2 1BA type for mechanical testing in one build. Each specimen has its own label to be able to follow part placement in terms of a set of coordinates in the X, Y, and Z axes. The strategy for part placement was set to be close to real manufacturing conditions. This means that parts are placed as close to each other as possible, but for the verification and validation of results, at least five parts were placed close to each other to minimize variations related to part placement (coordinates). All specimens for tensile testing were placed in four different orientations (see Figure 3).

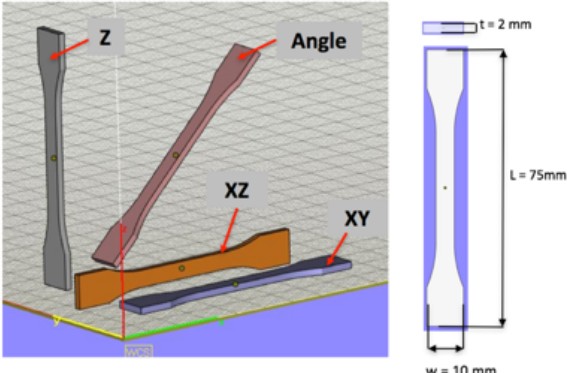

**Figure 3.** Schematic visualization of parts' orientations and dimensional features (where *t*—thickness, *w*—width and *L*—length).

All specimens were clustered into four groups based on the ISO/ASTM 52921 standard [36]. Each orientation group name was defined based on the size of each dimensional feature, from the highest to the smallest value:

- Group 1. XYZ (XY in Figure 3 )-oriented parts
- Group 2. XZY (XZ in Figure 3)-oriented parts
- Group 3. ZYX (Z in Figure 3)-oriented parts
- Group 4. Angle-oriented parts

By the Angle-oriented parts, the author means parts oriented at 45° between the X and Z axes.

Since the positioning of specimens was made based on the main requirement to fit in as many of them as possible, the number of standard specimens in each orientation group differs. Thus, 65 specimens are positioned in the XY orientation, 24 in the XZ orientation, 84 in the Z orientation, and 22 in the Angle orientation.

Among investigated features are STL model properties (surface, volume, and the number of mesh triangles), part orientation (angle for X/Y/Z-axes), and part placement (positioning) in the build chamber in terms of maximal, minimal, and central coordinates in the X-, Y-, and Z-axes. These features were extracted from the build layout prepared in Magics 22.03 software (see Figure 1).

*2.2. Conditioning of Specimens and Tensile Testing*

According to the DIN EN ISO 527-1, tensile testing was performed on universal test machine Zwick Z250 with one-week conditioning in the climate chamber at 70 °C and 62% RH. This is accelerated conditioning, which results in the same moisture content as mentioned in ISO 1110. In addition to one week of conditioning, specimens were kept for 1–2 days in the climate chamber at 23 °C and 50% RH before testing. The Zwick Z250 machine was loaded with 2.5 kN cell. The specimens were mounted in wedge grips with grip to grip distance set to 55 mm, and an initial gauge length of the extensometer of 25 mm.

*2.3. Application of Machine Learning Techniques to Predict Mechanical Properties without Prior Knowledge about Material Properties (PA)*

The successful prediction of mechanical properties of fabricated parts is an important factor of future adoption of additive manufacturing to produce end-user parts. The high-quality requirements and need for consistency are primary issues that have not been fully addressed yet. Typically, mathematical models need to be developed to make prior analysis of the mechanical properties possible. However, the complexity of the AM process leads to the simplification of the issue in one area in order to describe another one mathematically.

Therefore, another approach is required for obtaining robust predictions that will be less sensitive to unknown noise. Since there are examples of successful application of machine learning techniques in traditional manufacturing [32–35], these methods could be used to predict mechanical properties with limited information about the AM process itself.

Since the prediction of some output corresponds to regression task, linear regression models are compared with Gradient Boosting Regressor (GBR), Decision Tree Regressor, and AdaBoost Regressor machine learning techniques. The Python programming language was used to program these algorithms with the help of Scikit-Learn libraries [37], which consist of preprogrammed methods needed for this work. One of the advantages of using such techniques is a possibility to overcome the noise present and find patterns in the data that could not be identified with a "human eye".

However, these methods require a large number of data points for better performance. An additional challenge is a choice of correct techniques for the assigned tasks. In this work, the determination coefficient and mean square error (MSE) were considered as performance metrics for methods comparison. However, machine learning techniques often work as black boxes and models cannot be presented mathematically except for a description of the model's architecture. An additional requirement to the methods was the presence of a feature importance attribute. This information is needed for better understanding which features play important roles in the model development to estimate Tensile Modulus, Nomial Stress, and Elongation at break (or identify features that are in some way correlated to the mechanical properties).

Short Introduction to the Machine Learning Techniques Used

**Decision Tree Regressor** is a recursive algorithm that splits data into smaller subsets (separate classes) in order to form a tree, and it is important to choose the correct metrics for the best data split and to determine when a tree node should become a terminal.

Building a decision tree for regression is similar to classification, but the main difference is in using different metrics for the evaluation of the quality of the split (reduction of variance for regression task), and a leaf node is defined by numerical output. Optimization of these metrics will lead to the more stable performance of the algorithm.

However, when it comes to analysis of the large amount of data, this method has issues with scalability, stability, and robustness [29,38]. Another issue that should be addressed is an increase in complexity when large data samples are used. The total number of nodes, total number of leaves, tree depth, and number of attributes are metrics that can be controlled in order to minimize the complexity of decision tree [29]. Since these issues not always can be addressed, ensemble decision trees are used instead and are more robust.

**Gradient Boost Regressor** is an ensemble of decision trees. Instead of building one tree, this method predicts the desired outcome based on the additive regression model that uses decision trees as a weak learner [39]. Sequential fitting of a parameterized function (base learner) to current "pseudo"-residuals is done at each iteration by optimizing regression loss (e.g., least squares, absolute error) [40]. Friedman et al. [40] describe "pseudo"-residuals as minimization of the gradient of the loss function with respect to values of the regression model at each training data point for the current step.

Introduction of randomization in the process of training data set selection allows to improve accuracy and reduce the possibility of overfitting. This method of compiling a decision tree allows minimizing the errors at each next step, and therefore the boosting regressor is considered as a more reliable and robust method compared to the classic decision tree regressor.

**AdaBoost Regressor** (short for Adaptive Boost) is also an ensemble machine learning method. It works similarly to the Gradient Boost regressor, and the only difference is in the way weak learners are created at each iteration. Thus, AdaBoost changes the sample distribution at each iteration by changing the weights of each feature (the ones with the biggest error will have the highest weights).

Since all of the described methods need to be trained on the data before they can be used for a prediction task, the data should be divided into training and testing samples, usually with

a ratio of 75%/25% or 85%/15% when smaller data sets are used. For this purpose, there is a library *training_testing_split* in Scikit-Learn [37], which was used by the author in this work. The training data set is used to train the model (define the model), while the testing data set is needed for evaluating the model generalization. By comparing prediction accuracy obtained for training and testing data sets separately, one can understand whether under- or over-fitting are present. Overfitting is present when the model memorizes input data, while underfitting means that the model cannot find any patterns in the data due to a small number of data points. Both phenomena need to be overcome in order to obtain a robust model. In addition, optimization of hyperparameters of the models needs to be done in the future, and this task is out of the scope of this work.

The developed GBR and DTR machine learning models in this work have learning rate set as 0.1, the minimum number of samples in the leaf is set to 1, the minimum number of samples in the split is equal to 2, and the random state is set to 42. While the Gradient Boosting Regressor has the maximum depth set to 3 and criterion is chosen to be "Friedman MSE", DTR models do not have any limitation of max depth and model criterion is chosen to be the mean square error (MSE). Compared the other two models, the AdaBoost regressor is a more advanced version of the linear model, which has the same learning rate and random state as the two other types of models, but ABR models have a number of estimators equal to 50 with linear loss function. Even though described architectures are standard from the Scikit-Learn modules, the optimization of these models should be done when more variation in data is present, and, as mentioned above, model optimization is out of the scope of this study.

## 3. Results and Discussion

### 3.1. Description of the Collected Data

Mechanical properties depend on the part orientation, and this phenomenon is already presented in [12]. Therefore, the description of the mechanical properties should be done separately for each orientation. Four orientations were used in the current analysis, which are XYZ, XZY, ZYX, and Angle (45° between the X and Z axes) orientations. Tables 2–4 describe the data based on the Tukey range test (looking at percentiles), including standard deviation values.

**Table 2.** Statistical data evaluation for Tensile modulus for each orientation separately.

| Statistical Characteristics | XYZ | XZY | ZYX | Angle |
|---|---|---|---|---|
| std | 31.912 | 35.776 | **95.763** | 46.63 |
| mean | 1066.308 | 1051.951 | 958.25 | 1013.545 |
| 25% | 1046.308 | 1035.374 | 908.038 | 983.455 |
| 50% | 1067.77 | 1055.196 | 980.038 | 1010.291 |
| 75% | 1088.461 | 1074.403 | 1032.381 | 1043.599 |
| max | 1148.078 | 1112.222 | 1090.35 | 1118.547 |
| min | 968.483 | 933.376 | **648.079** | 907.983 |

**Table 3.** Statistical data evaluation for Nominal Stress for each orientation separately.

| Statistical Characteristics | XYZ | XZY | ZYX | Angle |
|---|---|---|---|---|
| std | 0.665 | 0.604 | **5.101** | 3.334 |
| mean | 37.341 | 35.429 | 22.031 | 30.773 |
| 25% | 36.918 | 35.019 | 18.303 | 28.478 |
| 50% | 37.476 | 35.491 | 22.19 | 29.91 |
| 75% | 37.832 | 35.809 | 25.826 | 31.383 |
| max | 39.186 | 36.576 | 32.241 | 37.744 |
| min | 35.519 | 34.132 | **10.09** | 26.219 |

**Table 4.** Statistical data evaluation for Elongation at break for each orientation separately.

| Statistical Characteristics | XYZ | XZY | ZYX | Angle |
|:---:|:---:|:---:|:---:|:---:|
| std | 0.727 | 0.679 | 1.096 | **2.311** |
| mean | 13.383 | 13.265 | 3.499 | 7.079 |
| 25% | 13.158 | 12.875 | 2.719 | 5.594 |
| 50% | 13.582 | 13.497 | 3.26 | 6.343 |
| 75% | 13.893 | 13.663 | 4.017 | 7.262 |
| max | 14.353 | 14.287 | 7.451 | 12.537 |
| min | 11.18 | 10.618 | **1.917** | 4.316 |

The results illustrated in Figure 4 support the anisotropic behaviour of PA 12 reported earlier [13,23]. While the Tensile moduli for the XYZ and XZY orientations have similar distributions, the tensile modulus for the ZYX orientation has the widest variance and the lowest value. Tensile modulus for Angle orientation has a narrow distribution, while the maximum value of Tensile modulus is 1065 MPa. In addition to variation between orientation, variation between the nominal value (provided by EOS for balanced machine settings for PA2200 material) and obtained results is also observed for Tensile modulus.

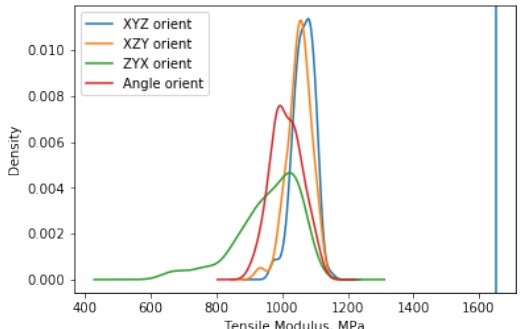

**Figure 4.** Distribution of tensile modulus for different orientations based on kernel distribution estimation (The straight line (1650 MPa) corresponds to the value from the EOS Balanced datasheet).

Similarly to the Tensile modulus, the distribution of nominal stress for different orientations is presented in Figure 5. The results of the XYZ and XZY orientations are similar to the Tensile modulus, although distribution for the ZYX orientation is much wider and the average value of nominal stress differs from the average value of the XYZ and XZY orientations. Additionally, if we compare the maximal value for the ZYX orientation with the value provided by EOS, one can observe that nominal stress is almost two times lower (from Table 3 it is ca. 22 MPa, while 42 MPa is expected) than that provided by EOS (dashed line in Figure 5). However, the value of nominal stress for Angle orientation is better than for ZYX. It is still relatively low compared with the results for the XYZ and XZY orientations.

Elongation at break compared with two other mechanical properties has more similar results to the ones provided by EOS, and is illustrated in Figure 6. According to the EOS datasheet, elongation at break is expected to be 4% for the ZYX orientation and 18% for the XYZ orientation. The average value of elongation at break for the ZYX and XYZ orientations are ca. 3.5% and 13.4%, respectively (taken from Table 4). Since there is no information from the EOS for other orientations, the observed values for the XZY and Angle orientations cannot be compared with nominal ones. However, it is expected the XZY orientation should have similar results to the XYZ orientation due to the dominating X-axis, and thus slicing of the specimens is performed perpendicularly to the direction of force load in a tensile test. Similarly, the ZYX and Angle orientations are sliced in such way that makes crack generation under the tensile test easier.

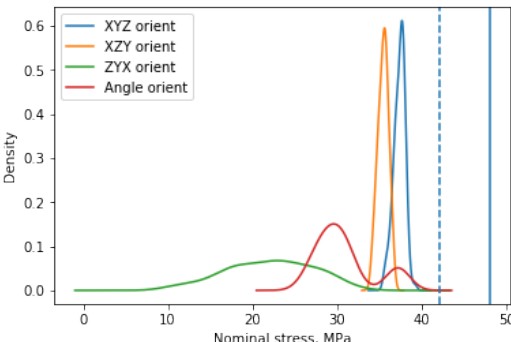

**Figure 5.** Distribution of nominal stress for different orientations based on kernel distribution estimation. The straight solid (48 MPa for the XYZ orientation) and dashed lines (42 MPa for the ZYX orientation) correspond to the values from the EOS Balanced datasheet.

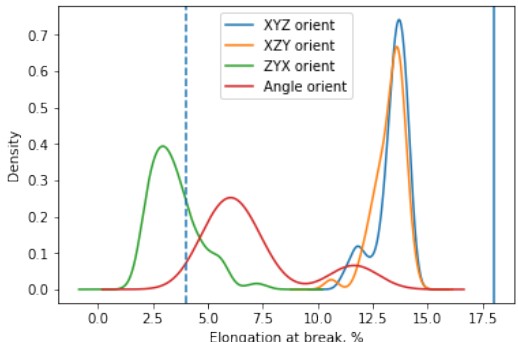

**Figure 6.** Distribution of elongation at break for different orientations based on kernel distribution estimation. The straight solid (18% for the XYZ orientation) and dashed lines (4% for the ZYX orientation) correspond to the values from the EOS Balanced datasheet.

The discussion on why this wide distribution is observed for the experimental results for the ZYX orientation will be provided in another article and is out of the scope of this article. In this work, different intelligent methods have been applied to discover the most important features, and obtained results are presented below.

### 3.2. Prediction of Tensile Modulus and Comparison of Models' Performances

The prediction of Tensile Modulus was performed for both different orientation groups and as one group without separation on the orientations (called "All" in Table 5). Results show that prediction is not yet possible in the XYZ, XZY, and Angle orientations due to various reasons. Since large data sets are a well-known requirement for the successful application of machine learning techniques, a small number of data points is one of the reasons why developed models for the XYZ, XZY, and Angle orientations do not work well. A limited number of investigated features could be considered as another reason for such model performance. Since many studies have earlier reported that material properties have a significant impact on mechanical properties [3,13,18,23,41], and in this study they are not considered, this may have an impact on the obtained results.

However, the prediction of Tensile Modulus based only on part positioning and STL model properties in the ZYX orientation has promising results. A model based on the gradient boost regressor algorithm outperformed other models with a prediction accuracy of 0.808 out of 1 (80.8% out of 100%) for the ZYX orientation. Additionally, it can be seen that an increased number of data points (see All in Table 5, where the number of data points represents number of data points used to train the model) leads to the higher prediction accuracy of 0.88 or 88% for the developed model. This means that the description of the part placement (positioning) in the build chamber in combination with part

orientation and STL model properties can be used to predict Tensile Modulus, especially when collected data is not divided in the groups corresponding to each orientation, but instead this information is used as an additional feature. This results show a similar trend to the one Caulfield et al. [12] have reported in their study. Thermal distribution in the build chamber has an impact on the resulting mechanical properties, and the presented results are in a good agreement with the results reported in [12]. Besides, more details on which parameters are used in this model are described in the section below.

**Table 5.** Prediction of Tensile Modulus with the help of machine learning techniques. Linear means linear regression models, GBR stands for Gradient Boost Regressor, DTR for Decision Tree Regressor, and ABR for AdaBoost Regressor.

| # Data Points | Orientation | Linear | | GBR | | DTR | | ABR | |
|---|---|---|---|---|---|---|---|---|---|
| | | $R^2$ | MSE | $R^2$ | MSE | $R^2$ | MSE | $R^2$ | MSE |
| 110 | XYZ | 0.129 | 602.945 | 0.272 | 503.781 | −0.76 | 1219.019 | 0.404 | 412.905 |
| 40 | XZY | −0.927 | 1421.147 | −2.297 | 2431.052 | −2.313 | 2442.847 | −2.102 | 2286.854 |
| 142 | ZYX | 0.48 | 4151.161 | **0.809** | **1524.994** | 0.764 | 1886.726 | 0.801 | 1586.234 |
| 74 | Angle | 0.03 | 1193.589 | −0.159 | 1426.052 | −0.469 | 1808.483 | 0.264 | 905.402 |
| 325 | All | 0.528 | 3876.678 | **0.888** | **916.721** | 0.819 | 1488.903 | 0.879 | 995.159 |

### 3.3. Prediction of Nomial Stress and Comparison of Models' Performances

The prediction of Nominal stress (maximal nominal stress) was also performed based only on the part positioning in the build chamber and STL model properties (number of mesh triangles, surface, and volume). However, compared to results obtained for Tensile Modulus, a number of successful models for prediction of Nominal Stress have higher prediction accuracies. While models in the XYZ and XZY orientations show similar behavior as for Tensile Modulus, the ZYX and Angle orientations have even better prediction accuracy (0.902 and 0.906, respectively).

In addition, an interesting phenomenon can be observed for Angle orientation. Since a number of data points for this orientation are smaller than in the XYZ orientation, model performance is significantly better. This can be explained by looking at the linear regression models. In this case, the prediction of Nominal Stress in the Angle orientation is even possible with the use of a linear model, which has a relatively high prediction accuracy—0.867 out of 1. This means that a combination of part positioning and STL model properties has a linear correlation with Nominal Stress in Angle orientation. Therefore, the application of more advanced methods results in improved model performance with higher prediction accuracy (like AdaBoost regressor in Table 6). The prediction of Nominal Stress in the ZYX orientation has similar results to Tensile Modulus, and the best model is based on the GradientBoost Regressor.

**Table 6.** Prediction of Nominal Stress with the help of machine learning techniques. Linear means linear regression models, GBR stands for Gradient Boost Regressor, DTR for Decision Tree Regressor, and ABR for AdaBoost Regressor.

| # Data Points | Orientation | Linear | | GBR | | DTR | | ABR | |
|---|---|---|---|---|---|---|---|---|---|
| | | $R^2$ | MSE | $R^2$ | MSE | $R^2$ | MSE | $R^2$ | MSE |
| 110 | XYZ | 0.393 | 2.093 | 0.197 | 2.768 | 0.173 | 2.852 | 0.372 | 2.165 |
| 40 | XZY | −0.645 | 0.827 | −0.013 | 0.509 | −0.043 | 0.525 | 0.50 | 0.251 |
| 142 | ZYX | 0.252 | 15.983 | **0.902** | **2.102** | 0.893 | 2.293 | 0.843 | 3.356 |
| 74 | Angle | 0.867 | 0.527 | 0.855 | 0.578 | 0.839 | 0.642 | **0.906** | **0.373** |
| 325 | All | 0.763 | 10.193 | **0.964** | **1.537** | 0.963 | 1.609 | 0.937 | 2.689 |

As can be seen in Table 6, the best model with the highest prediction accuracy and the lowest MSE corresponds to the case when gathered data is not separated into orientation-based groups. Besides, orientation is considered as an additional feature, similarly to the previously reported results for Tensile Modulus. The prediction accuracy of Gradient Boosting regressor model is 0.964 out of 1, and this means that we can already use this model for the prediction of Nominal Stress for a similar type of parts.

However, if there is a need for prediction of Nominal stress value for a specific orientation group, the prediction can be done for the ZYX and Angle orientations at this moment. In order to improve models in other orientations, additional information on the material, as well as more data points, are required. The author will address these issues in the future.

The prediction of Nominal Stress in Angle orientation is possible by using any of the described models in Table 6. However, when higher accuracy is required, the AdaBoost model would be the best choice. Otherwise, when a mathematical model is needed, then linear regression model can be used:

$$
\begin{aligned}
y = -832.16 * x_1 - 6.819 * x_2 + 415.9 * x_3 + 13.639 * x_4 + \\
+ 416.24 * x_5 - -6.819 * x_6 - 7.44e - 04 * x_7 + \\
+ 0.823 * x_8 + 0.197 * x_9
\end{aligned}
\tag{1}
$$

where $y$ is the Nominal Stress in Angle orientation, $x_1$ is the central cordinate $x$, $x_2$ is central coordinate $y$, $x_3$—min coordinate $x$, $x_4$—min coordinate y, $x_5$—max coordinate $x$, $x_6$—max coordinate $y$, $x_7$—max corrdianate $z$, $x_8$—volume, and $x_9$—surface.

### 3.4. Prediction of Elongation at Break and Comparison of Models' Performances

Similarly to trends observed for Nominal Stress in Angle orientation, the elongation at break in Angle orientation also has a linear correlation with the investigated features. As can be seen in Table 7, models in the XYZ, XZY, and ZYX orientations have relatively low prediction accuracies and therefore cannot be used at this moment. This phenomenon can be explained by looking at the previously published results [12], which state that material properties have a significant impact on mechanical properties, including elongation at break. In order to improve developed models, material properties in terms of viscosity and virgin/used powder ratio should be determined.

However, when data is not separated on the orientation-based groups, prediction accuracy increases significantly. Thus, one can assume that the number of data points is an important factor. Besides, AdaBoost Regressor has outperformed all other algorithms with a prediction accuracy 0.987 out of 1 with relatively low MSE comparing with other methods.

**Table 7.** Prediction of Elongation at break with a help of machine learning techniques. Linear means linear regression models, GBR stands for Gradient Boost Regressor, DTR—Decision Tree Regressor, and ABR—AdaBoost Regressor.

| # Data Points | Orientation | Linear | | GBR | | DTR | | ABR | |
|---|---|---|---|---|---|---|---|---|---|
| | | $R^2$ | MSE | $R^2$ | MSE | $R^2$ | MSE | $R^2$ | MSE |
| 110 | XYZ | 0.261 | 0.0907 | 0.0146 | 0.121 | −0.168 | 0.143 | 0.165 | 0.103 |
| 40 | XZY | 0.476 | 0.667 | 0.46 | 0.681 | 0.453 | 0.696 | 0.486 | 0.655 |
| 142 | ZYX | 0.269 | 0.963 | **0.67** | **0.434** | 0.587 | 0.545 | 0.638 | 0.476 |
| 74 | Angle | 0.889 | 0.500 | 0.795 | 0.927 | 0.749 | 1.129 | **0.903** | **0.439** |
| 325 | All | 0.965 | 0.739 | **0.987** | **0.284** | 0.985 | 0.326 | 0.971 | 0.616 |

Since the Linear regression model in the Angle orientation has relatively high prediction accuracy (0.889 out of 1), this model can be used in cases when the mathematical description is required:

$$y_{eb} = 49.54 * x_1 - 2.757 * x_2 - 25.02 * x_3 + 5.516 * x_4 -$$
$$- 9.45e - 06 * x_5 - 24.52 * x_6 - 2.757 * x_7 + \quad (2)$$
$$+ 2.497e - 03 * x_8 - 2.41e - 02 * x_9$$

where $y_{eb}$ is the Elongation at break in the Angle orientation, $x_1$ is the central cordinate x, $x_2$ is central coordinate y, $x_3$—min coordinate x, $x_4$—min coordinate y, $x_5$—min coordinate z, $x_6$—max coordinate x, $x_7$—max corrdianate y, $x_8$—max coordinate z, and $x_9$—surface.

### 3.5. Feature Importance for Prediction of Tensile Modulus

Prediction of Tensile modulus with model prediction accuracy higher than 0.8 out of 1 was considered as successful. However, when the author attempted to develop a model for each orientation separately, models for the XYZ, XZY, and Angle orientations appeared to be unsatisfactory, and therefore it is not possible to extract a robust evaluation of which features are important. One of the reasons why this is not possible at this moment is related to the number of data points (which is much lower comparing with ZYX orientation and all data together). It is a well known fact that the more data points one has, the better the performance of machine learning techniques. Additionally, information about material properties is crucial for mechanical properties and adding this information in the future may improve models significantly.

Since evaluation of the relative importance of the features on Tensile Modulus is possible only for the ZYX orientation and without data separation on orientation-groups, namely All orientations, Figure 7 illustrates the differences observed for these two groups. According to the results shown in Figure 7a, one of the STL model properties, which is volume, is depicted as relatively the most significant feature for the non-linear model based on the gradient boost regressor in the ZYX orientation.

Since it is known from EOS that volume and surface parameters help to define energy concentration for a specific part, more energy will be concentrated in the build chamber where a part is placed. Thus, parts with larger values of volume (which means that the part is larger) will have larger energy concentration in the build chamber. In addition to this feature, part placement will also play an important role. Since it was already previously reported that in polymer powder bed fusion system temperature distribution within a build chamber is different, and also influences the mechanical properties [12,20], it can be assumed that coordinates in the X, Y, and Z-axes help to identify regions where specimens are placed. Besides, in combination with the volume parameter, they provide a much better description of energy concentration and temperature distribution in the specific areas of the build chamber. In addition, machine learning techniques have a unique possibility of determining hidden patterns between feature and output, which are not visible to the "human eye", and therefore can provide better results.

In order to better understand why coordinates in the X and Z-axes are listed in the top 5 of the most important features for the ZYX orientation, visualization of the Tensile modulus as a function of X and Z coordinates is needed. As Figure 8 (sum of relative importance for all features is equal to 1) shows, when central coordinates in the X-axis are larger than 250 mm, the tensile modulus has the lowest values. This indicates the correlation between coordinates in the X-axis and Tensile Modulus. Since part positioning in the build chamber is always described as a combination of all three axes, the coordinates in the Y and Z-axes provide additional information on how the Tensile Modulus changes depending on the position in the build chamber.

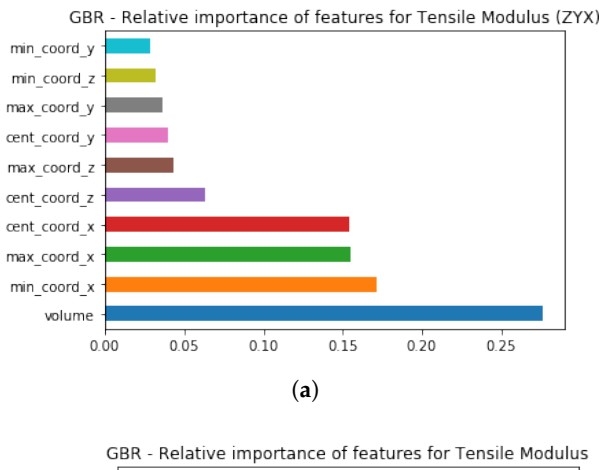

(**a**)

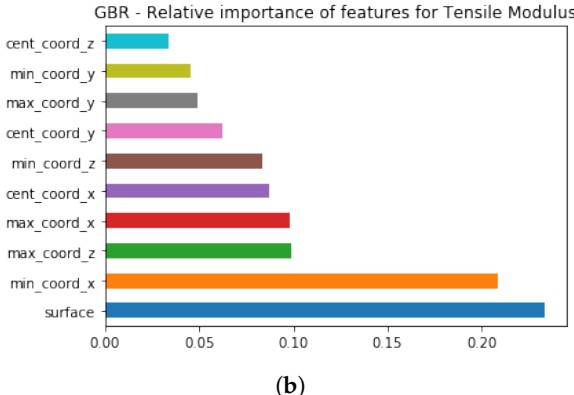

(**b**)

**Figure 7.** Feature relative importance based on the specimen orientations (**a**) Tensile Modulus—ZYX orientation. (**b**) Tensile Modulus—All orientations.

However, when all orientations are analyzed as one data set, the list of the most important features differs from the one proposed for the ZYX orientation. Another STL model property is listed as relatively the most important, which is surface, and as it has been already mentioned earlier, the surface feature, similarly to volume, describes in more detail the energy concentration for a specific part, and thus contributes to the description of temperature distribution for a specific area in the build chamber. The coordinates follow the surface feature but in a different order. While minimal coordinate X is second in the list, its relative importance is a bit higher than for the ZYX orientation. Additionally, maximal coordinates in the Z and X-axes are important to the same extent.

The difference in the order of the coordinates in different axes compared with the feature importance list proposed for the ZYX orientation could be caused by the number of parts in each orientation group and their placement in the build chamber. For example, the most significant number of specimens is in the the ZYX orientation (168 from two builds), and XYZ is the second largest group (130 specimens from two builds). Therefore, one can assume that results for the XYZ orientation group will significantly influence on the importance of coordinates in the X, Y, and Z-axes. Another assumption could be made for the description of temperature distribution in the build chamber. In other words, including parts from other areas in the build chamber allows defining patterns regarding values of Tensile Modulus, and thus leads to a better prediction accuracy of the model.

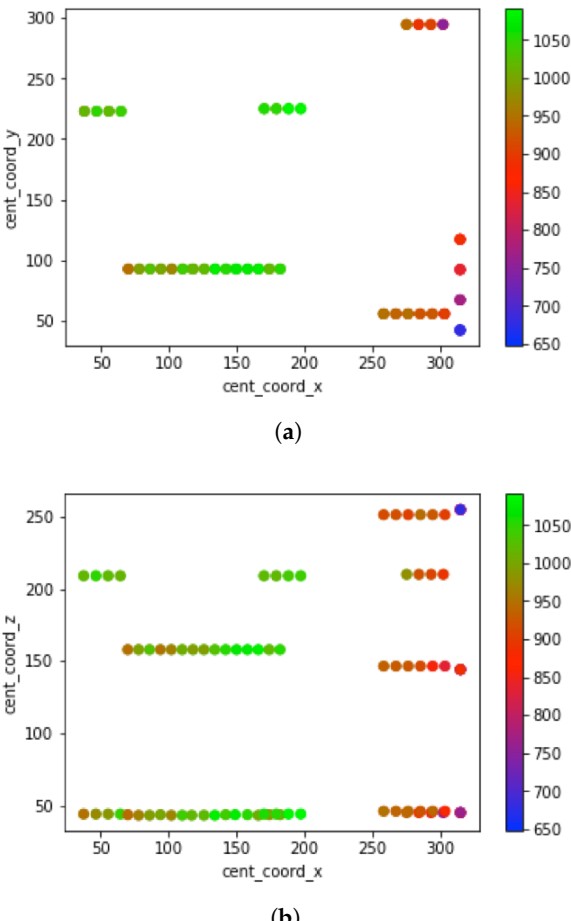

**Figure 8.** Visualization of Tensile Modulus for ZYX orientation (**a**) As a function of central coordinates in the X- and Y-axes in mm. (**b**) As a function of central coordinates in the X- and Z-axes in mm.

### 3.6. Feature Importance for Prediction of Nominal Stress

The Nominal Stress can be predicted for the ZYX, Angle, and All orientations with a higher prediction accuracy (0.902, 0.906, and 0.964 respectively) than for Tensile Modulus. For example, the list of the features based on the relative importance in the ZYX orientation for Nominal stress is different (see Figure 9a) from the list proposed for Tensile Modulus. The Gradient Boost Regressor algorithm has weighted number of mesh triangles as the most important feature, which is followed by such features as surface, central, and minimal coordinates in the X-axis, number of mesh triangles, and volume, in the listed order.

The positioning of the specimens is also highlighted by the algorithm in terms of the coordinates in the X-axis and maximal coordinate in the Z-axis. Since the connection between STL model properties and part positioning has been described for Tensile Modulus as a possible way of defining temperature distribution in the build chamber, with special attention to the energy concentration, similar trends are observed in models developed to predict Nominal Stress. The number of mesh triangles influences the way a part is sliced on the layer, which results in the construction of parts' contours at each layer for an additive machine, and thus contributes to the energy concentration for each specific part. This assumption is in good agreement with previously published studies [42,43].

However, when it comes to the analysis of features' importance for the developed model in Angle orientation, STL model properties are still in the list of being important, but their contribution is less significant compared to the ZYX/All orientations (see Figure 9b). One of the reasons why volume and surface features are among the least significant for the developed model can be the distribution of actual values for all specimens in Angle orientation. Thus, surface values are in the

range 1401.787–1432.563 mm$^2$, while the values range for volume is even smaller (1029.925–1035.427 mm$^3$). However, the value range for a surface feature for all specimens (without separation on orientations) has a significant difference (e.g., 1381.555–1441.187 mm$^2$) compared with volume values (1029.925–1035.427 mm$^3$).

However, comparison of these ranges raises another question on why such a small range for the volume feature is important for prediction of both Tensile Modulus and Nominal Stress in the ZYX orientation. In order to be able to answer this question, more experimental work is needed to be done in the future, where the variation of the STL model properties should be of main concern. Another assumption could be made by looking at the prediction accuracy for the linear model, which is equal to 0.867 out of 1. This could mean that the linear correlation between investigated features and Nominal Stress is present for angle orientation, while the correlation between STL model properties and Nominal Stress is absent.

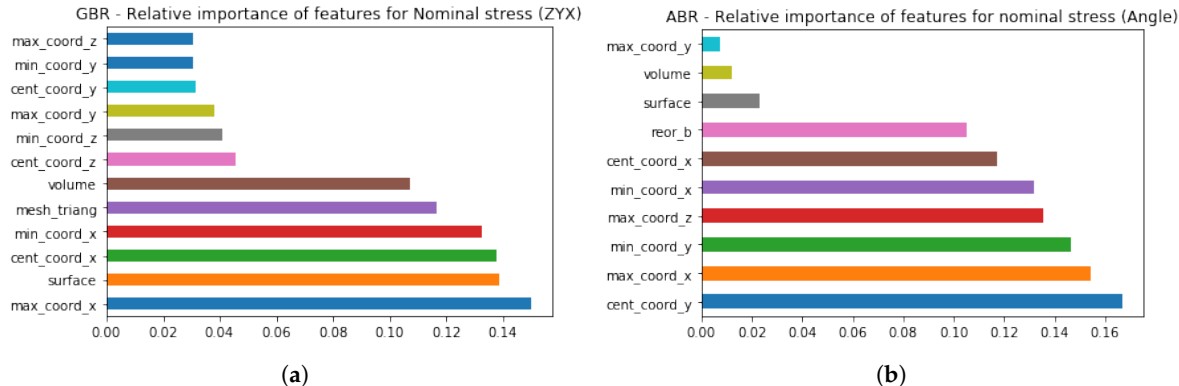

(**a**)                    (**b**)

**Figure 9.** Feature relative importance based on the specimen orientations (**a**) Nominal stress—ZYX orientation. (**b**) Nominal Stress—Angle orientation.

Results obtained when all orientations are analyzed together are illustrated in Figure 10. As can be seen, specimens' positions in the X-axis is listed as the most important feature, and special attention is paid to the minimal coordinate in the X-axis. In order to understand better why minimal coordinate in the X-axis is chosen to be more important than, for instance, central or maximal coordinates, there is a need for visualizing how Nominal Stress is dependent on these coordinates. Since the graphical representations for central and maximal coordinates in X-axis looks alike, the minimal coordinate in the X-axis is compared only with central coordinate X, and their comparison is shown in Figure 11. Even though Figure 11a has similar dependencies compared with Figure 11b, one can still observe the better defined correlation between minimal coordinate X and Nominal Stress. Even though at this moment it is not possible to confirm the proposed assumptions, it is also important to highlight that feature importance in this work is described from the perspective of developed models instead of making conclusions whether these parameters influence the values of mechanical properties.

In other words, if another model with high accuracy is proposed, the list of the most important features will be different, and it also differs depending on the machine learning techniques used to develop these models. Besides, the main difference between the techniques lies in the different approaches of weighting the input parameters. However, analyzing feature importance for models can help in this study to raise new questions and show a new direction for the experimental work in the future.

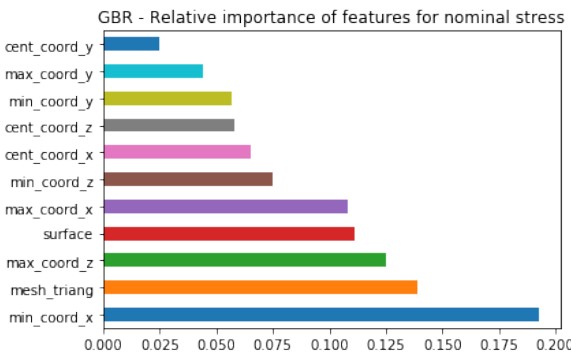

**Figure 10.** Feature relative importance for Nominal Stress—All orientations.

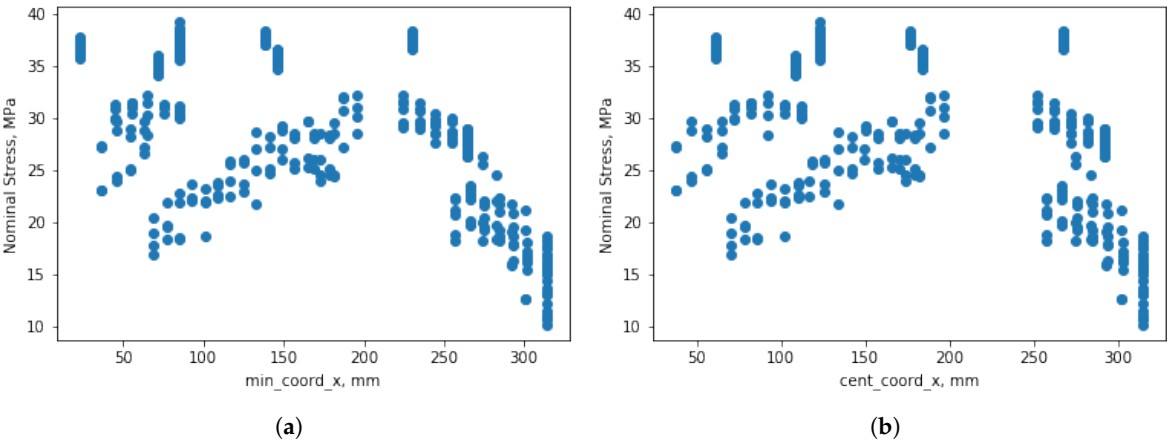

(**a**)    (**b**)

**Figure 11.** Comparisson of the correlations between Nominal Stress and coordinates in the X-axis (for all data) (**a**) Nominal Stress vs minimal coordinate in the X-axis. (**b**) Nominal Stress vs central coordinate in the x axis.

### 3.7. Feature Importance for Prediction of Elongation at Break

The features that described orientation angle had dominated over other parameters and has negatively influenced the model robustness for models developed to predict elongation at break, especially when all orientation groups were joined. Therefore, it was decided by the author to develop models based on the specimens' placements in the build chamber and STL models, and exclude orientation angle from the feature's list. This has resulted in lower prediction accuracy but has helped to overcome an issue of overfitting (memorized data instead of defined patterns and dependencies). However, it is still important to keep in mind that part orientation has the most significant impact on the elongation at break (see Figure 6, where raw values are in the range 1.917%–33.715%).

Prediction of elongation of the break has higher prediction accuracy in Angle orientation (0.903 out of 1) than in the ZYX orientation (0.67 out of 1), even though the number of data points for ZYX orientation is almost two times larger. While it seems that a number of data points are not an important factor for machine learning techniques, there are other reasons that explain the observed phenomenon. Development of a robust model with good generalization requires not only many data points but also features that influence the parameters which the model is predicting. Therefore, in future studies, there is a need for introducing material properties and additional information that will explain temperature distribution in the build chamber in a more detailed way. Additionally, it is important to analyze linear model performance in Angle orientation because this model also has prediction accuracy of 0.889, which could mean that Elongation at break has a linear correlation with the positioning of the specimens in the build chamber similarly to the Nominal Stress.

Figure 12 illustrates that the most significant parameter is the minimum coordinate in the X-axis. This feature is important for the model, but it does not mean that the minimum X coordinate influences the value of elongation at break. However, the positioning of specimens in the build chamber allows identifying changes in the temperatures in the build chamber, which influences mechanical properties, similar to other reports [12].

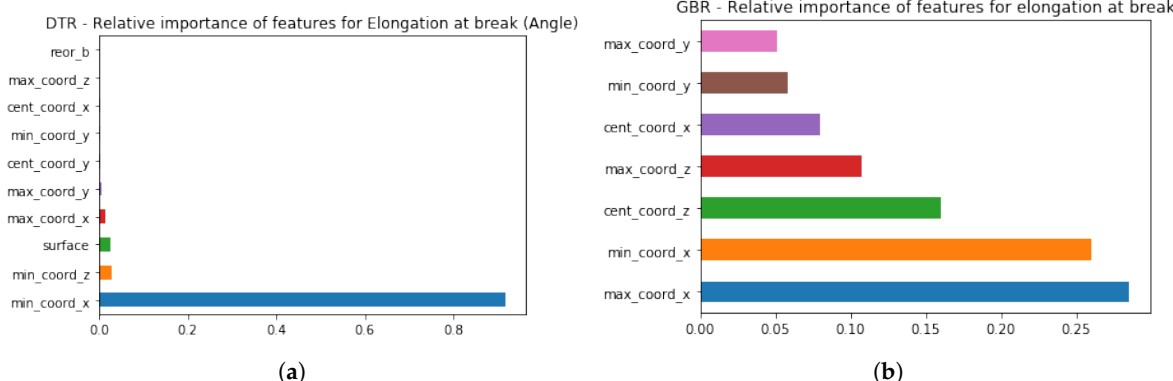

**Figure 12.** Relative importance of the features based on the specimen orientations (**a**) Elongation at break—Angle orientation. (**b**) Elongation at break—All orientations.

Additionally, it is important to mention that the speed under which force was loaded on the specimens also has an impact on the values of elongation at break. Since the difference between values mentioned in EOS datasheets and obtained in the experimental work are similar, one can assume that this factor can be neglected.

While Elongation at break for Angle orientation could be predicted based on four features (see Figure 12a, when all orientations are joined into one dataset, the number of important features for the model is different (see Figure 12b). As can be seen from Figure 12b based only on the positioning of the specimens in the build chamber, there is a high potential in predicting elongation at break. Similarly to Tensile Modulus and Nominal Stress, coordinates in the X- and Z-axes are defined as the most important ones. These results could be different for the AM machine, and therefore, in the future work, more data needs to be collected based on experimental work, which will be focused on the moving specimens within build chamber, with special attention to the features listed as the most important. Since the author could not find any similar reports in order to compare the obtained results, verification and validation of these results will be done in the future based on the additional experiments.

## 4. Conclusions

Additive manufacturing (AM) is an attractive technology for the manufacturing industry due to flexibility in design and functionality, but inconsistency in quality is one of the major limitations that does not allow utilizing this technology for producing end-use parts. Mechanical properties are dependent on both processes and material parameters of an additive manufacturing process, but there is a limited number of studies that focus on the STL model properties and part positioning in the build chamber as features that can be used to describe and predict mechanical properties in addition to material and process parameters.

Since both material and process parameters were kept the same for the two builds analyzed in this study, this information is neglected, and the main focus is on the analysis of build layout. From Figures 4–6, it can be seen that the repeatability of the two builds is good, and this means that control of material properties can help to avoid inconsistency in the quality that usually appears from build to build. However, distributions in the ZYX orientation for Tensile Modulus and Nominal Stress are wide, which has not been reported earlier, and therefore, this issue should be analyzed in more detail in another article.

When it comes to the prediction of mechanical properties by using traditional methods like linear regression, it can be seen from Tables 5–7 that prediction accuracy for linear regression (except for Nominal Stress and Elongation at break in Angle orientation) is low, and therefore more advanced techniques need to be used. Since machine learning techniques have also been widely used in traditional manufacturing [32–35], these methods were applied in this study to develop non-linear prediction models.

These techniques are chosen for the reason that machine learning methods have an ability to find hidden patterns in the data that could be structured or unstructured (which is not possible with classical methods like DOEs), and then use those patterns to make predictions for unknown problems [9]. Therefore, data presented in different studies can be used together with data collected from practical experiments. Another benefit of using machine learning techniques lies in the long-term use of these models. In other words, the introduction of new data from each use of an additive manufacturing system will help to achieve continuous improvements of the model's performance. The models that are pre-trained based on the data collected from experimental work, like the one presented in this study, can be used as a starting point by companies, and then these models should be adjusted for specific a type of product.

However, machine learning techniques require large datasets for better performance, and this trend can be observed based on the different model accuracies for orientation-based modeling and when all orientations are joined into one dataset. Models developed on the largest datasets ("All orientation" in Tables 5–7) have the highest prediction accuracy and, therefore, have potential to be used for the prediction of mechanical properties for similar types of parts. When it comes to the prediction of the mechanical properties for each orientation group individually, Tensile Modulus could be predicted only in the ZYX orientation, Nominal Stress could be predicted in the ZYX and Angle orientations, and Elongation at break could be predicted in Angle orientation. All other models require additional information about the material and increase the number of specimens that are analyzed.

Since two builds were identical, there is a need for introducing more variations into the data such as; material properties, changing specimens' positions and orientations, producing specimens with other sizes and designs, and changing the values of STL model properties. In addition, more attention should be paid to the energy description for each part based on their placement in the build chamber.

**Funding:** This research was funded by Norges Forskningsråd as a part of MKRAM project.

**Conflicts of Interest:** The author declares no conflict of interest.

## Abbreviations

The following abbreviations are used in this manuscript:

| | |
|---|---|
| DOE | Desifn of Experiment |
| MSE | Mean Square Error |
| GBR | Gradient Boosting Regressor |
| ABR | AdaBoost Regressor |
| DTR | Decision Tree Regressor |

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
