# Peer review of "Application of Machine Learning Techniques to Predict the Mechanical Properties of Polyamide 2200 (PA12) in Additive Manufacturing"

_applsci, doi:10.3390/app9061060_

Reviewer 1 Report

The work presents the use of three different machine learning algorithms as applied to SLS processing, as well as a linear regression model. The authors should include a bit more detail on the method of implementation of the three machine learning algorithms in Python. For example, what parameters were selected in each model utilised. It would have been nice to see that the process parameters were chose according to a Design of Experiments (DoE) in order to allow comparison to DoE also. However, the comparison between the three machine learning algorithms and linear regression model is also interesting. 

The details of the geometry features in figure 7 to 12 need to be presented and explained. Some of these results could be left out without effecting the overall conclusions from the paper. The paper is overly long in this sense and could be reduced in lenght here.

Author Response

The author would like to thank you for the valuable feedback. Detailed information on what has been changed in the article is described in the file attached below.

Reviewer 2 Report

Regarding the introduction part:

During the introduction the authors review a considerable amount of literature. Which is appropriate and relevant to the study presented.

However, author should include literature related to classical:

(1) Design of Experiments (DOEs) on mechanical characterization based on energy density and process parameters studies for polymers. For example the following literature can be included and discussed:

Flores Ituarte, I., Wiikinkoski, O., Jansson, A., 2018. Additive Manufacturing of Polypropylene: A Screening Design of Experiment Using Laser-Based Powder Bed Fusion. Polymers (Basel). 10, 1293. https://doi.org/10.3390/polym1012129

(2) Part orientation, powder morphology and machine process parameters have been also studied in structured DOE. There is a detailed discussion on how powder distribution affects mechanical performance; however, there is no explanation on how this can be overcome. Complementary research also explains how part positioning in the build chamber as well as stl and machine technology affect geometrical part accuracy. For example the following literature can be included and discussed:

Flores Ituarte, I., Huotilainen, E., Wiikinkoski, O., Tuomi, J., 2018. Experiment with Machine Vision for Polymer Flowability Analysis in Powder Bed Fusion, in: Chua, C.K., Yeong, W.Y., Tan, M.J., Liu, E.J. (Eds.), 3rd International Conference on Progress in Additive Manufacturing (Pro-AM 2018). Singapore.

Flores Ituarte, I., Coatanea, E., Salmi, M., Tuomi, J., Partanen, J., 2015. Additive Manufacturing in Production: A Study Case Applying Technical Requirements. Phys. Procedia 78, 357–366. https://doi.org/10.1016/j.phpro.2015.11.050 

Regarding the research gap:

Authors should discuss and explain briefly using the included literature what is the real benefit of using novel techniques based on machine learning versus classical DOE methods such as surface responses methods or fractional design of experiments to characterize mechanical performance of SLS technologies.

Both approach can generate linear and non-linear regression models including process parameter interactions. In the case of stcrutyred and planned DOE it is also possible to understand and model the phenomena as a “white box”

Why Machine learning offers benefits in this case?

This could be interesting to briefly explain in the intro and discuss deeper in the conclusions section.

Materials and methods:

 (1) I have some difficulties to understand the process parameters involved in the manufacturing of both batches. Could you explain in detail, what are the process parameters (e.g. laser power, scanning speed, layer thickness, etc.) involved in the fabrication of both “Runs”? Could you explain the printing mode? Laser type of the machine and all the necessary data to replicate the experiment? This could be included in the Table 1.

(2) How do you measure for what you call “STL model properties (number of mesh triangles, surface and volume)”.The STL file quality might have an stronger impact if rounded surfaces and curvatures. A tensile testing specimens is rather simple with only curves in the neck. Did you account for tensile specimens that broke in the neck as a part of the results? If yes, can you specify how many of the tested ones failed that way and if this was correlated with the fact of lower STL quality.

Results and Discussion:

I would rather structure the paper slightly different to help readability

(1) I would rather combine 3. Results and 4. Discussion into the same section

(2) Take subchapter 3.2 and 3.2.1 to the methodology part in section 2

(4) Present 3.2.2 as a part of the results and discussion and structure the new 3. Results and Discussion section as:

3.1. Description of the collected data

3.2. Prediction of Tensile Modulus and comparison of models’ performance

3.3. Prediction of Nomial Stress and comparison of models’ performan

3.4. Prediction of Elongation at break and comparison of models’ performanc

3.5. Feature importance for prediction of Tensile Modulus

3.6. etc…

Conclusions:

I kind of miss a discussion presenting what is the real benefit of machine learning methods to model mechanical performance of SLS parts.

The research is valuable and creative; however, I would like that the author discuss and explain what is the real benefit of using novel techniques based on machine learning versus classical DOE methods such as surface responses methods or fractional design of experiments to characterize mechanical performance of SLS technologies.

Both approach can generate linear and non-linear regression models including process parameter interactions. In the case of structured and planned DOE it is also possible to understand and model the phenomena as a “white box”

Why Machine learning offers benefits in this case? Can we used unstructured data from scientific publications to be process with machine learning? Why do you plan a time consuming experiment with 2 x 358 samples if you later are planning to model it as a black box?

Typos:

Sentence 54: Srtess?

Sentence 78: wo agist?

Sentence 467: orientation?

 I would rather check the language for more typos using proof reading services.

Author Response

The author would like to thank you for the valuable feedback. The detailed information about changes in the manuscript is described in the attached document.
